# The Off-Label Use of Inhaled Nitric Oxide as a Rescue Therapy in Neonates with Refractory Hypoxemic Respiratory Failure: Therapeutic Response and Risk Factors for Mortality

**DOI:** 10.3390/jcm8081113

**Published:** 2019-07-27

**Authors:** Hsiu-Feng Hsiao, Mei-Chin Yang, Mei-Yin Lai, Shih-Ming Chu, Hsuan-Rong Huang, Ming-Chou Chiang, Ren-Huei Fu, Jen-Fu Hsu, Ming-Horng Tsai

**Affiliations:** 1Department of Respiratory Therapy, Chang Gung Memorial Hospital, Linkou, Taiwan; 2Department of Respiratory Therapy, Chang Gung University, Taoyuan, Taiwan; 3Division of Neonatology, Department of Pediatrics, Chang Gung Memorial Hospital, Taoyuan, Taiwan; 4College of Medicine, Chang Gung University, Taoyuan, Taiwan; 5Division of Neonatology and Pediatric Hematology/Oncology, Department of Pediatrics, Chang Gung Memorial Hospital, Yunlin, Taiwan

**Keywords:** Inhaled nitric oxide, pulmonary hypertension, off-label use, premature infants, mortality

## Abstract

Objectives: The indication of inhaled nitric oxide (iNO) used in preterm infants has not been well defined. Neonates with refractory hypoxemia may benefit from the pulmonary vasodilatory effects of iNO. The aim of this study was to investigate the off-label use of iNO as a rescue therapy. Methods: Between January 2010 and December 2017, all neonates who received iNO as a rescue therapy from a tertiary-level medical center were enrolled, and those who were not diagnosed with persistent pulmonary hypertension of newborn (PPHN) were defined as having received off-label use of iNO. The controls were 636 neonates with severe respiratory failure requiring high-frequency oscillatory ventilation but no iNO. Results: A total of 206 neonates who received iNO as a rescue therapy were identified, and 84 (40.8%) had off-label use. The median (interquartile) gestational age was 30.5 (26.3–37.0) weeks. Neonates receiving iNO had significantly more severe respiratory failure and a higher oxygenation index than the controls (*p* < 0.001). Respiratory distress syndrome and secondary pulmonary hypertension after severe bronchopulmonary dysplasia (BPD) were the most common causes of the off-label iNO prescription. Of the 84 neonates with off-label use of iNO, 53 (63.1%) had initial improvement in oxygenation, but 44 (52.4%) eventually died. The overall mortality rate was 41.7% (86/206). After multivariate logistic regression, extremely preterm (odds ratio [OR] 5.51; *p* < 0.001), presence of pulmonary hemorrhage (OR 2.51; *p* = 0.036) and severe hypotension (OR 2.78; *p* = 0.008) were the independent risk factors for final mortality. Conclusions: iNO is applicable to be an off-label rescue therapy for premature neonates with refractory hypoxemia due to severe pulmonary hypertension and bronchopulmonary dysplasia.

## 1. Introduction

Over two-thirds of neonates in the neonatal intensive care unit (NICU) require mechanical ventilation due to respiratory failure, and approximately 12–25% of them had refractory hypoxemia [1,2,3]. Inhaled nitric oxide (iNO) has been documented to be effective for the treatment of persistent pulmonary hypertension of newborn (PPHN) and hypoxic respiratory failure in near-term and term-born infants [4,5,6,7,8]. In recent years, iNO has been increasingly prescribed to neonates less than 34 weeks of gestation with respiratory distress syndrome (RDS) [8,9,10,11,12]. These studies have demonstrated that the improved pulmonary outcomes in preterm neonates were as effective as that in late preterm and term-born neonates without negative impact on survival [8,9,10,11,12].

Some studies have documented the effectiveness of iNO in improving oxygenation and rescuing extremely preterm infants with severe RDS [10,12,13,14]. However, other studies have found that iNO is not associated with reduced in-hospital mortality or survival without bronchopulmonary dysplasia (BPD) among these critically ill neonates [11,15,16]. Furthermore, both the National Institutes of Health (NIH) [17] and the American Academy of Pediatrics (AAP) [18] suggest that iNO should not be used as an early-rescue therapy in preterm infants due to safety concerns, including an increased risk of intraventricular hemorrhage (IVH). The application of iNO has extended from term and near-term neonates with PPHN to extremely premature infants with severe RDS or BPD based on the anti-inflammatory and pulmonary vasodilatory effects of iNO [8,9,10,11,12,13,14,15,16]. However, only a small percentage of preterm neonates with RDS have concurrent PPHN physiology [19]. Most neonates with severe BPD were only clinically presumed pulmonary hypertension. These “off-label uses” of iNO have not been well investigated. In this study, we aimed to test the efficacy and safety of the off-label use of iNO in neonates without a definite diagnosis of PPHN.

## 2. Materials and Methods

### 2.1. Study Design, Setting and Data Source

We performed this cohort study using a deidentified dataset approved by the Institutional Review Board of Chang Gung Memorial Hospital (CGMH) with a waiver of informed consent. Linkou CGMH is the largest tertiary-level medical center in Taiwan, with the NICU comprising 3 units and 107 beds. The average annual admission to the NICU of Linkou CGMH is 800 patients, and most of them are inborn preterm infants. All ventilator data including mode, settings, and inhalation therapies have been prospectively recorded by well-trained respiratory therapists for over a decade. Baseline demographics, perinatal factors and conditions, chronic comorbidities, diagnoses of respiratory diseases, treatments, and outcomes were retrospectively collected from electronic records and validated. In order to characterize the efficacy of the off-label use of iNO, we categorized those with iNO use during hospitalization into the standard-use group (for PPHN, irrespective of the underlying etiology) and the off-label use group (for clinically presumed pulmonary hypertension or diagnoses other than PPHN). In the off-label use group, we further classified them as early use of iNO (less than 7 days old) and late use of iNO (over 7 days old). Besides, we enrolled a control group of neonates who required high-frequency oscillatory ventilation (HFOV) due to severe respiratory failure and/or those with poor response to conventional ventilations. The severity of illness was assessed by the Neonatal Therapeutic Intervention Scoring System (NTISS) [20], which was prospectively recorded in our NICU. This study was an 8-year study between January 2010 and December 2017. The datasets used/or analyzed during the current study are available from the corresponding author on reasonable request.

The policy of iNO usage in the NICU of Linkou CGMH includes the standard rescue therapy for refractory hypoxemia caused by PPHN, which was documented by echocardiography reports or oxygen saturation differences of over 5–10% or partial pressure of oxygen (PaO2) differences of 10–20 mmHg between lower limbs and right upper limb before initiation of iNO. The primary diagnoses attributable to PPHN include neonatal sepsis, pulmonary hemorrhage, congenital diaphragmatic hernia (CDH), hydrops fetalis, idiopathic PPHN, meconium aspiration syndrome, and idiopathic/possible RDS. Other uses of iNO will be considered as “final rescue” when these neonates have refractory hypoxemia with poor response to all therapeutic strategies, including HFOV, surfactant therapy, muscle relaxant or sedation, and cardiac inotropic agents. These patients did not have a definite diagnosis of PPHN and the use of iNO was entirely at the discretion of the treating neonatologist. For iNO usage on neonates with severe BPD combined with secondary pulmonary hypertension documented by echocardiography, we also considered this as “off-label use” because these patients were extremely preterm and iNO was usually used later on one to two month olds.

In our institute, most infants had a starting dose of iNO of 20 parts per million (ppm) and rarely 10 ppm. If no clinical response to hypoxemia was observed, the dose of iNO would increase to 40 ppm with a maximum of 80 ppm. The serum levels of methemoglobin and blood gas analysis were routinely checked before iNO and followed at 4, 8, 12 and 24 h after initiation of iNO and were then checked daily. The blood gas analysis and ventilatory settings of HFOV were routinely recorded every 4 h in the most critical course or when needed. The attending physician determined the initiation of iNO, dose, associated therapeutic strategies, and the decision to maintain, wean, or discontinue iNO.

### 2.2. Study Outcomes

The primary outcome was in-hospital mortality, including those who were critically ill and discharged at the family’s request and transferred to other hospitals. Secondary outcomes were BPD and survival without BPD, defined as requiring supplemental oxygen or respiratory support at 36 weeks of corrected gestational age [21]. The iNO treatment responses were as follows [10,12,22]: (1) good response: a fraction of inspired oxygen (FiO2) reduction ≧ 20% within 12 h post iNO administration and iNO can be successfully weaned within 3 days; (2) partial response: improved to a certain extent, but still needed iNO and HFOV for over 72 h; (3) partial response and then failure: improved to a certain extent, but still needed iNO and HFOV for over 72 h and the treatment eventually failed; and (4) failure: failed to improve on HFOV with/without iNO treatment and the patient died later or were sent for extracorporeal membrane oxygenation (ECMO) treatment within 72 h.

### 2.3. Statistical Analyses

Intergroup comparison was performed using the chi square test or Fischer’s exact test for categorical variables and using Student’s t-test or Wilcoxon rank-sum test for continuous variables as appropriate. All *p* values were two tailed, and *p* values of <0.05 were considered to be statistically significant. Logistic regression methods were used to analyze the risk factors for final in-hospital mortality in neonates who received iNO. Statistical analyses were performed using SPSS version 21.0 (SPSS^®^, Chicago, IL, USA).

## 3. Results

During the study period, a total of 206 neonates received iNO during their hospitalization in our NICU. Among them, 122 were in the standard-use group and 84 were in the off-label use group. Of the 84 patients in the off-label use group, 57 received iNO within the first 7 days of life. We enrolled a total of 636 critically ill neonates with respiratory failure requiring HFOV for rescue therapy as the control group. The baseline demographics of the study and control groups are summarized in Table 1. For all neonates with PPHN, 59.2% (122/206) received iNO as the rescue therapy. In all neonates with iNO use, the median (interquartile [IQR]) gestational age and birth weight were 30.5 (26.3–37.0) weeks and 1319.5 (789.3–2675.0) g. Of the neonates receiving iNO therapy, 154 (74.8%) were preterm infants, and 96 (46.6%) were less than 30 weeks of gestation. Of the 206 patients with iNO use, 162 (78.6%) had initiation of iNO within the first 3 days of life. The percentage of IVH (≥ grade III) in infants who received iNO therapy was 9.2% (19/206), which was comparable to that in the control group (8.0%). The median (IQR) duration of iNO use was 5.0 (3.0–11.0) days, and the median duration of intubation with mechanical ventilation was 17.0 (7.0–59.3) days.

In the off-label use group, neonates who received iNO within the first week of life were significantly more prematurely born than neonates with PPHN and received iNO therapy (28.4 ± 5.4 weeks vs. 34.3 ± 5.5 weeks, *p* < 0.001), with severe RDS (66.7%) and sepsis (26.3%) being the most common primary diagnoses. However, for neonates with off-label use of iNO after 7 days old, the most common diagnoses were severe BPD (70.4%) and secondary pulmonary hypertension (51.9%). While the neonates with sepsis accounted for one-third (33.3%) of the iNO off-label use group, nearly half of the neonates in each group had perinatal distress (low Apgar score ≦ 7 at five minutes) (Table 1). Besides, the gestational age and birth body weight of the control group were comparable to those of the early off-label use group (less than 7 days old when iNO was used).

Table 2 summarizes the initial ventilator settings and treatment modalities for the study and control groups, who had refractory respiratory failure requiring iNO or HFOV. Neonates with iNO use for PPHN or those with off-label use of iNO had significantly more severe hypoxemia, higher oxygenation index, higher AaDO2, higher mean airway pressure and more hypercabnia when compared with the neonates in the control group (all *p* values < 0.05). Over 70% of the neonates with iNO use had an oxygenation index (OI) greater than 20 at initiation of iNO, and over 75% of them started using iNO within 48 h after initiation of HFOV, indicating that these patients did not respond to HFOV under the maximum settings. Of the patients with iNO use, 81.1% used cardiac inotropic agents, and 54.2% required more than one cardiac inotropic agent. The use of cardiac inotropic agents in the study group (especially in neonates with PPHN) was significantly more common than that in the control group (94.3% vs. 72.2%, *p* < 0.001).

We evaluated the response and improvement of oxygenation in the study group within the first 48 h after initiation of iNO, and physiologic variables pertinent to the degree and onset of hypoxemia are displayed in Table 3. For the control group, the initiation of HFOV was set as day 0. Of the 206 patients with iNO treatment, 143 (69.4%) had an initial response and significant improvement of oxygenation in the first 24 h. However, only 76 (36.9%) were able to have iNO and HFOV high ventilator settings weaned within 3 days, 67 (32.5%) needed iNO for over 3 days, and 36 (17.5%) died within the first 72 h. Of the 143 patients who had an initial response to iNO therapy, 18 patients eventually died of other underlying comorbidities. The overall in-hospital mortality rate of the cohort with iNO therapy was 41.7% (n = 86), which was relatively higher than that of the control group (vs. 31.4%, *p* = 0.072). Besides, late iNO treatment (after 7 days of life) in the off-label use group had the worst response rate and the highest rate of in-hospital mortality (all *p* value < 0.001 by χ2 test with Bonferroni correction and log rank test) (Figure 1).

Compared with the control group, neonates with iNO as a rescue therapy had significantly higher illness severity (judged by the NTISS scores), higher oxygenation index at initiation of iNO and/or HFOV, and higher OI during the first day and second day after initiation of iNO (Table 3). The control group also had better therapeutic response than the standard-use group and the off-label use group. The controls had significantly greater tendency to progress to BPD than those with early iNO use (53.9% vs. 32.4%, *p* < 0.001), and had a comparable in-hospital mortality rate when compared with the standard-use group. Furthermore, neonates with PPHN treated with iNO had a significantly higher rate of survival without BPD than the controls (72.5% vs. 42.1%, *p* < 0.001).

We also investigated the risk factors for final in-hospital mortality in neonates with iNO treatment (Table 4). In univeriate regression analysis, extremely preterm (GA < 30 weeks), presences of pulmonary hemorrhage, primary diagnosis other than PPHN, severe hypotension (defined as those requiring two or more cardiac inotropic agents or use of epinephrine), sepsis, perinatal asphyxia (Apgar score < 7 at five minutes) and secondary pulmonary hypertension were identified to be independently associated with final mortality in neonates with iNO. The initial oxygenation index at the initiation of iNO was significantly associated with response to iNO therapy (*p* = 0.001 for every increase of 10 in OI), but was not associated with final in-hospital mortality, indicating that those with good initial iNO responses may die of other chronic comorbidities or recurrent infections during prolonged hospitalization. In multivariate logistic regression analysis, extremely preterm (odds ratio [OR] 5.51; 95% confidence interval [CI] 2.54–11.93, *p* < 0.001), presence of pulmonary hemorrhage (OR 2.51; 95% CI 1.06–5.92, *p* = 0.036), and severe hypotension (OR 2.78; 95% CI 1.31–5.94, *p* = 0.008) were the independent risk factors for final mortality.

## 4. Discussion

In animal studies of hypoxia pulmonary injury, iNO has been found to have anti-inflammatory effects, promotion of angiogenesis, apoptosis inhibition and anti-oxidative injury [23,24,25]. The beneficial effects of iNO on extremely preterm infants have been increasingly investigated [10,11,12,13,14,16,18,26,27,28,29,30]. Recently, the American Heart Association, American Thoracic Society and Pediatric Pulmonary Hypertension Network have considered iNO to be superior to other pulmonary vasodilators in the treatment of preterm infants with severe pulmonary hypertension [31,32]. Therefore, we defined the use of iNO in hypoxemic neonates with diagnoses other than PPHN as “off-label rescue use”. In contrast to most of the published randomized controlled trials that focused on infants less than 34 weeks of gestation and early use of iNO during the few days of life [10,11,12,13,14,16,26,27,28], we were the first to investigate the rescue use of iNO in the late treatment of neonates who already had BPD and secondary pulmonary hypertension. We found that the rescue use of iNO improved initial oxygenation in over half of the preterm neonates with refractory hypoxemia.

Our institute did not have a specific protocol for the initiation of rescue therapy. The choice of treatment modalities was completely at physicians’ discretion. In addition, the use of iNO could be limited by the unavailability of machines for NO delivery at that time. Therefore, 41% (*n* = 84) cases with PPHN did not receive iNO, and the study cohort reflected the physicians’ preference. From a clinical perspective, iNO was utilized in the most severe and critically ill neonates in our institute. Although the existing evidence does not support the use of iNO as rescue regimens in the care of infants less than 34 weeks of gestation [17,18,33], most of our off-label iNO prescription were used the final rescue and most subjects had an initial high OI over 25 and poor responses to all therapeutic strategies. Based on our results, we still considered iNO to be the potential final rescue given that ECMO is not possible for extremely preterm infants.

Recent studies have also demonstrated that a significant proportion of preterm infants with severe hypoxemia can be rescued by iNO therapy during their critical stage, especially during the first 3 days of life [10,18,22,26,27,28]. The better response of exogenous iNO administration in preterm infants in the early life may be due to the underlying mechanism of transient deficiency of endogenous nitric oxide generation, especially in those with prolonged preterm rupture of membranes [14,34]. However, iNO was found less effective in patients receiving it after 7 days old, and inflammation may be an important factor because sepsis, recurrent pneumonia, and chronic lung diseases are the dominant etiologies of late-onset acute pulmonary hypertension. Therefore, patients who had early use of iNO have been reported to have better effects than those who received iNO after 7 days old.

In our cohort, we found that the response to iNO was mostly within the first 24 h after iNO initiation, and a good response of late off-label use of iNO (after 7 days old) also occurred within the first 6 h. Therefore, most previous studies defined iNO responders as neonates with a FiO2 reduction of over 20% within the first 3 to 6 h after iNO initiation [10,12], and the iNO responders had a significantly higher survival rate than the non-responders [10,12]. Long-term exposure (25 days) and cumulative effects of iNO were found to have a modest but statistically significant beneficial effect in a randomized controlled trial [30], but the study subjects were 7–21 days of age. In our cohort, 96 of 206 (46.6%) neonates had iNO treatment for over one week and 41 (19.9%) had iNO treatment for over 2 weeks. Most of these patients with long-term iNO exposure were late treatment due to severe BPD and secondary pulmonary hypertension; therefore, only 34 (35.4%) of these patients survived.

We found the presence of pulmonary hemorrhage, severe hypotension, and extremely preterm birth to be independently associated with final mortality, rather than the severity of hypoxia or higher OI at the initiation of iNO. These findings suggested that using iNO as a rescue therapy for most severe hypoxemia cases can be effective, and underlying pulmonary physiology or comorbidities are important contributors to final mortality. Although the case mortality rate was high up to 41%, not all of these patients died of poor response and refractory respiratory failure. We also confirmed that the surviving responders had a low morbidity rate, which is consistent with previous studies conducted on iNO-treated term born neonates [24].

This study has some important limitations. First, it lacked a definite protocol for iNO use in neonates, and all therapeutic strategies were determined by the attending physicians. We were unable to confirm the optimization of the combination of iNO and HFOV according to the severity of hypoxemia. Second, the demographics of the control group and the study group did not fully match. However, it is almost impossible to conduct a placebo-controlled trial in this cohort due to high illness severity, the occasional unavailability of iNO and HFOV, and ethics issues. Clinicians must do their upmost to treat these critically ill neonates. Third, this cohort came from a single center in a single country. The results of this study are inevitably less generalizable to the whole world. The main advantage of this study is its originality, as all conditions of iNO use were enrolled into the analysis and the adequate sample size was adequate even though the occurrence rate of these critically ill conditions are low.

## Figures and Tables

**Figure 1 jcm-08-01113-f001:**
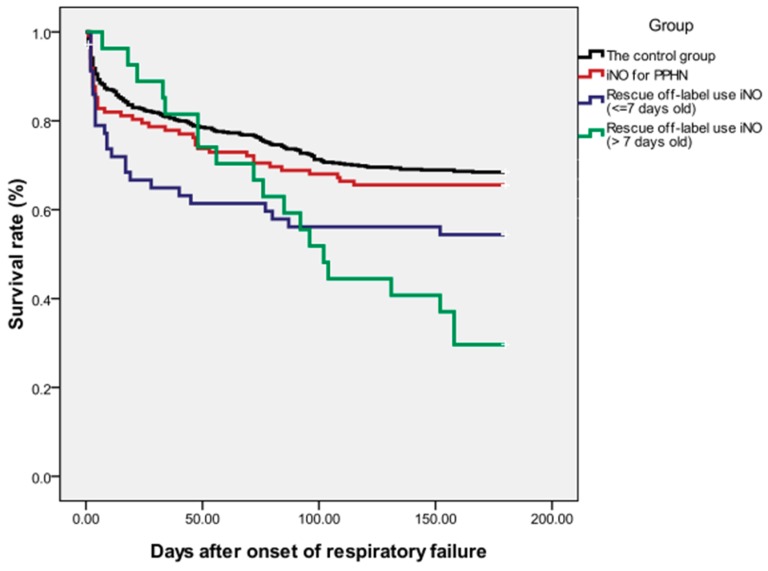
Survival following onset of respiratory failure from 842 neonates in the neonatal intensive care unit of CGMH 2010-2017. The Kaplan-Meier graph is stratified by the study and control groups. (iNO: inhaled nitric oxide, PPHN: persistent hypertension of newborn).

**Table 1 jcm-08-01113-t001:** The baseline demographics and primary pulmonary diagnoses of neonates with inhaled nitric oxide (iNO) use and the control group.

Baseline Demographics	Neonates with iNO Use			The Control Group (All Neonates with Respiratory Failure Requiring HFOV)
	Standard Rescue (for PPHN)	Off-Label Use of iNO as a Rescue Therapy		
		iNO Within First 7 Days	iNO after 7 Days	
Case Number (*n*)	122	57	27	636
Gestational age (week), median (IQR)	34.6 (27.0–37.6)	28.4 (25.0–35.3)	30.5 (26.0–36.5)	27.6 (25.3–31.6)
Birth body weight (g), median (IQR)	2057 (845–2945)	1150 (714.5–2462.5)	1305 (788.5–2532)	940.0 (721–1545)
Gender (male/female), *n* (%)	79 (64.8)/43 (35.2)	30 (52.6)/27 (47.4)	21 (77.8)/6(22.2)	384 (60.4)/252 (39.6)
Apgar score 5 min ≦ 7, *n* (%)	63 (51.6)	27 (47.4)	15 (55.6)	360 (56.6)
Primary pulmonary diagnosis *, *n* (%)				
PPHN	122 (100)	0 (0)	0 (0)	84 (13.2)
Respiratory distress syndrome	54 (44.3)	38 (66.7)	5 (18.5)	411 (64.6)
Congenital diaphragmatic hernia	8 (6.6)	1 (1.8)	0 (0)	9 (1.4)
Meconium aspiration syndrome	13 (10.7)	0 (0)	0 (0)	21 (3.3)
Sepsis	17 (13.9)	15 (26.3)	13 (48.1)	120 (18.9)
Pneumonia	4 (3.3)	0 (0)	1 (3.7)	22 (3.5)
Pneumonthorax	17 (13.9)	6 (10.5)	0 (0)	77 (12.1)
Pulmonary hemorrhage	15 (12.3)	5 (8.8)	0 (0)	65 (10.2)
Hydrops fetalis	5 (4.1)	0 (0)	0 (0)	20 (3.1)
Bronchopulmonary dysplasia	0 (0)	0 (0)	19 (70.4)	28 (4.4)
Secondary pulmonary hypertension	0 (0)	0 (0)	14 (51.9)	5 (0.8)
Other concurrent comorbidities *				
Congenital heart disease ^&^	2 (1.6)	6 (10.5)	2 (7.4)	15 (2.4)
Intraventricular hemorrhage ≧ grade III	10 (8.2)	8 (14.0)	1 (3.7)	51 (8.0)
Congenital gastrointestinal obstruction	1 (0.8)	0 (0)	3 (11.1)	20 (3.1)

PPHN: persistent pulmonary hypertension of newborn; HFOV: high-frequency oscillatory ventilation; IQR: interquartile range. * Primary pulmonary diagnoses and concurrent comorbidities indicated the presence of disease entities when the iNO or HFOV was initiated, not including the subsequent outcomes of respiratory failure. ^&^ Indicated cyanotic congenital heart disease or symptomatic acyanotic heart disease, not including patent ductus arteriosus.

**Table 2 jcm-08-01113-t002:** Initial ventilator setting and other treatment modalities of neonates when inhaled nitric oxide (iNO) use was initiated and the control group when high-frequency oscillatory ventilation was initiated.

Ventilation Setting When on iNO or When on HFOV	Neonates with iNO Use			The Control Group (All Neonates with Respiratory Failure Requiring HFOV)
	Standard Rescue (for PPHN)	Off-Label Use of iNO as a Rescue Therapy		
		iNO Within First 7 Days	iNO After 7 Days	
DeltaP, cm H_2_O	33.5 (24.8–100.0)	38.0 (30.0–100.0)	75.0 (35.0–100.0)	100.0 (32.0–100.0)
Frequency, Hz	12.0 (11.0–13.0)	12.0 (11.0–13.0)	11.0 (11.0–13.0)	12.0 (11.0–13.0)
FiO_2_	100 (80.0–100.0)	100 (80.0–100.0)	90.0 (70.0–100.0)	80.0 (60.0–100.0)
Oxygenation index	21.0 (11.0–40.5)	26.0 (13.5–40.0)	28.0 (14.0–44.0) *	18.0 (10.0–34.0)
AaDO_2_	553.5 (385.5–593.0)	544.0 (392.5–608.0)	463.0 (290.0–596.0)	390.0 (243.5–567.8)
Mean airway pressure (cm H_2_O)	14.0 (12.0–15.3)	13.0 (11.0–15.5)	14.0 (11.0–18.0)	11.0 (10.0–14.0)
PH *	7.22 (7.09–7.29)	7.22 (7.02–7.34)	7.37 (7.34–7.43)	7.25 (7.14–7.36)
PaO_2_	47.6 (32.8–66.8)	42.0 (28.0–70.5)	47.6 (32.8–66.8)	54.7 (39.3–82.0)
PaCO_2_	54.2 (44.0–68.3)	54.0 (40.5–73.5)	48.0 (40.3–60.5)	52.0 (42.0–65.9)
Other treatment modalities				
Surfactant use	79 (64.8)	37 (64.9)	1 (3.7)	389 (61.2)
Dopamine	115 (94.3) **	42 (73.7)	10 (37.0)	459 (72.2)
Dobutamine	81 (66.4) **	23 (40.4)	6 (22.2)	222 (34.9)
Epinephrine	22 (18.0)	12 (21.1)	3 (11.1)	48 (7.5)
Milrinone	26 (21.3)	7 (12.3)	6 (22.2)	32 (5.0)
NTISS score	21.0 (18.0–24.0)	21.0 (17.5–24.0)	21.5 (18.0–24.0)	19.5 (16.0–22.0) *

The ventilator settings were expressed as median (interquartile). Data are number (percentage). * Some data may be checked after sodium bicarbonate replacement. All values were compared with each other with * *p* < 0.05 and ** *p* < 0.001 by *χ*^2^ test after Bonferroni correction. NTISS score: The Neonatal Therapeutic Intervention Scoring System.

**Table 3 jcm-08-01113-t003:** Treatment response of patients in the study group (neonates with inhaled nitric oxide [iNO] use) and the control group during the first three days of iNO and HFOV use and the final outcomes.

Oxygenation Index after Initiation of iNO or HFOV (the Control Group)	Neonates with iNO Use (the Study Group)			The Control Group (All Neonates with Respiratory Failure Requiring HFOV)
	Standard Rescue (for PPHN)	Off-Label Use of iNO as a Rescue Therapy		
		iNO Within First 7 Days	iNO After 7 Days	
Oxygenation index, median (IQR)				
Within the first 2 h	25.0 (13.0–39.3)	30.0 (13.0–48.5)	28.0 (15.0–44.0)	14.0 (8.0–28.0)
6 h later	17.5 (10.0–24.3)	19.0 (9.5–30.0)	18.0 (8.0–43.0)	10.0 (6.0–17.0)
12–24 h later ^&^	15.0 (8.0–22.5)	15.0 (6.0–30.5)	18.0 (9.0–44.0) **	8.0 (5.0–15.0)
2 days later ^&^	11.0 (5.0–27.0)	18.0 (7.0–33.5)	20.0 (6.0–47.0) **	8.0 (4.0–19.0)
Treatment response, *n* (%)				
Good response	50 (41.0)	22 (38.6)	4 (14.8) **	419 (65.9)
Partial response	33 (27.0)	11 (19.3)	10 (37.0)	76 (11.9)
Partial response and then failure	7 (5.7)	3 (5.3)	3 (11.1)	45 (7.1)
Failure	32 (26.2)	21 (36.8)	10 (37.0)	96 (15.1)
Final outcomes, *n* (%)				
Progress to BPD	37 (30.3)	21 (36.8)	24 (88.9) **	343 (53.9)
Survival without BPD	58/80 (72.5) **	16/32 (50.0)	0/8 (0) **	183/435 (42.1)
In-hospital mortality	42 (34.4)	25 (43.9)	19 (70.4) **	200 (31.4)

All values were compared with each other with * *p* < 0.05 and ** *p* < 0.001 by *χ*^2^ test after Bonferroni correction. The worst conditions during the period were taken. ^&^ Some patients died of respiratory failure and were not considered.

**Table 4 jcm-08-01113-t004:** Multivariate logistic regression to identify the independent risk factor for final mortality in neonates with iNO therapy.

Variables	Univariate Logistic Regression		Multivariate Logistic Regression	
	OR (95% CI)	*p* Value	OR (95% CI)	*p* Value
Gestational age (GA)				
≧37 weeks	1 (Reference)	-	1 (Reference)	-
30–36 weeks	3.02 (1.21–7.57)	0.018	3.16 (1.35–7.42)	0.008
<30 weeks	5.16 (2.33–11.43)	<0.001	5.51 (2.54–11.93)	<0.001
Low apgar score at 5 min (<7)	1.88 (1.18–3.02)	0.009	1.22 (0.70–2.11)	0.485
Initial oxygenation index				
0–10	1 (Reference)	-		
11–20	0.65 (0.25–1.69)	0.379		
21–30	1.17 (0.49–2.81)	0.726		
31–40	1.08 (0.40–2.83)	0.897		
>40	1.86 (0.80–4.29)	0.148		
Pulmonary hemorrhage	2.72 (1.09–6.81)	0.032	2.51 (1.06–5.92)	0.036
Sepsis	1.81 (0.94–3.55)	0.077	1.32 (0.66–2.65)	0.439
Severe hypotension *	2.41 (1.16–4.97)	0.018	2.78 (1.31–5.94)	0.008
Secondary pulmonary hypertension	5.21 (1.50–16.56)	0.019	6.79 (0.81–15.05)	0.078
Diagnosis other than PPHN (off-label use)	2.02 (1.14–3.56)	0.015	1.60 (0.80–3.20)	0.184

* Defined as neonates requiring two or more cardiac inostropic agents or use of epinephrine. PPHN: persistent pulmonary hypertension of newborn; OR: odds ratio; 95% CI: 95% confidence interval.

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
