# Peer review of "The Off-Label Use of Inhaled Nitric Oxide as a Rescue Therapy in Neonates with Refractory Hypoxemic Respiratory Failure: Therapeutic Response and Risk Factors for Mortality"

_jcm, 2019, doi:10.3390/jcm8081113_

Round 1
Reviewer 1 Report
Rescue off label use of inhaled nitric oxide in neonates with refractory hypoxemic respiratory failure: therapeutic response and risk factors of mortality – Hsiao et al.
Subjects: Between January 2010 and December 2017, all neonates from a tertiary-level medical center who received rescue iNO were enrolled, and those with diagnoses other than persistent pulmonary hypertension of newborn (PPHN) were defined as “rescue off-label use”.
Respiratory distress syndrome and secondary pulmonary hypertension after severe bronchopulmonary dysplasia (BPD) were the most common causes of the rescue iNO off-label prescription. 53 (63.1%) of 84 neonates with iNO off-label use had initial improvement of oxygenation, but 44 (52.4%) finally died. The overall mortality rate was 41.7% (86/206). After multivariate logistic regression, extremely preterm (odds ratio [OR] 5.51; P < 0.001), presence of pulmonary hemorrhage (OR 2.51; P=0.036) and severe hypotension (OR 2.78; P=0.008) were the independent risk factors for final mortality.
The authors conclude: Rescue iNO can be the last choice in premature neonates with refractory hypoxemia due to severe pulmonary hypertension and bronchopulmonary dysplasia.
Comments:
Inhaled nitric oxide use in premature neonates continues to be a controversial topic. The abstract is confusing. Objectives are not clear.
Those with diagnoses other than persistent pulmonary hypertension of newborn (PPHN) were defined as “rescue off-label use” – kindly explain.
“These studies demonstrated improved pulmonary 46 outcomes in preterm neonates as effectively as in late preterm and term neonates without negative impact on survival [8-12]” – Kindly explain. Every retrospective study post NIH statement has shown that off-label iNO use is detrimental.
The authors have to define iNO rescue therapy in the manuscript.
In this study, we aimed to test the effectiveness and safety of iNO off-label use in neonates without definite diagnoses of PPHN – in Taiwan, was iNO use deemed off-label?
Materials & methods: The iNO practical policy in the NICUs of Linkou CGMH included the standard rescue therapy for refractory hypoxemia due to PPHN, which was documented by echocardiography reports or 88 oxygen saturation differences of > 5-10% or PaO2 differences of 10-20 mmHg between lower limbs and right upper limb before initiation of iNO.
The median (IQR) median duration of iNO use was 5.0 (3.0-11.0) days – kindly explain –how did this timeframe include infants with BPD?
However, the most common diagnosis of neonates with iNO off-label use after one week old were severe BPD (70.4%) and secondary pulmonary hypertension (51.9%) – Please explain. This is contradicting the above statement.
Besides, the control group had comparable gestational age and birth body weight with those in the early use of iNO off-label use group. In a retrospective study, what does comparable mean? Comparable is not a scientific term in a retrospective study and make the study weak.
Neonates with iNO use for PPHN or those with off label use of iNO had significantly more severe hypoxemia, higher oxygenation index, higher AaDO2, higher mean airway pressure and more hypercabnia when compared with the control group (all P values < 0.05). – American Thoracic Society recommends that CO2 levels normalize before iNO initiation. Kindly comment.
In neonates with iNO use, more than 70% of patients had an OI greater than 20 at initiation of iNO, and more than 75% of patients had initiation of iNO within 48 hours after initiation of HFOV, which indicated that these patients did not respond to HFOV under the maximum settings. – How will they respond if there exists respiratory acidosis with hybercarbia?
The authors cited come papers – response to iNO is not merely change in FiO2 (expressed as percentage not decimal), some papers have mentioned it as fraction of change in PaO2 to FiO2.
Discussion needs to be focused. Was years of us added as a variable?
Author Response
RE: JCM-524850
Rescue off label use of inhaled nitric oxide in neonates with refractory hypoxemic respiratory failure: therapeutic response and risk factors of mortality
Journal of Clinical Medicine
Dear Editor,
Thank you for your appreciated comments on our manuscript. We had the manuscript revised, all according to the reviewers’ and editor’s suggestions. We underline every change and highlight in red color on the revised manuscript. The replies for the reviewers’ criticisms are as followings. We hope this revised version can be acceptable.
Best regards,
Ming-Horng Tsai
Chief, Division of Neonatology and Pediatric Hematology/Oncology, Department of Pediatrics, Yunlin Chang Gung Memorial Hospital,Taiwan, R.O.C.
Comments from Reviewer No.1 :
Subjects: Between January 2010 and December 2017, all neonates from a tertiary-level medical center who received rescue iNO were enrolled, and those with diagnoses other than persistent pulmonary hypertension of newborn (PPHN) were defined as “rescue off-label use”.
Respiratory distress syndrome and secondary pulmonary hypertension after severe bronchopulmonary dysplasia (BPD) were the most common causes of the rescue iNO off-label prescription. 53 (63.1%) of 84 neonates with iNO off-label use had initial improvement of oxygenation, but 44 (52.4%) finally died. The overall mortality rate was 41.7% (86/206). After multivariate logistic regression, extremely preterm (odds ratio [OR] 5.51; P < 0.001), presence of pulmonary hemorrhage (OR 2.51; P=0.036) and severe hypotension (OR 2.78; P=0.008) were the independent risk factors for final mortality.
The authors conclude: Rescue iNO can be the last choice in premature neonates with refractory hypoxemia due to severe pulmonary hypertension and bronchopulmonary dysplasia.
Comments:
Inhaled nitric oxide use in premature neonates continues to be a controversial topic. The abstract is confusing. Objectives are not clear.
Those with diagnoses other than persistent pulmonary hypertension of newborn (PPHN) were defined as “rescue off-label use” – kindly explain.
Reply:
Thank you for your instructive advice. The term “rescue off-label use” will make readers confused, so I use the term “off-label use of iNO as a rescue therapy” in the revised manuscript. Because use of iNO in near-term and term born infants with PPHN is the FDA approval standard treatment, clinical use of iNO in preterm infants with diagnosis other than PPHN is usually considered as “off-label use of iNO” in the literature. I will revise as the following: The objective of this study is to investigate the off-label use of iNO as a rescue therapy in the NICU (The last sentence of the objectives in the abstract, also the title of this manuscript is revised).
“These studies demonstrated improved pulmonary 46 outcomes in preterm neonates as effectively as in late preterm and term neonates without negative impact on survival [8-12]” – Kindly explain. Every retrospective study post NIH statement has shown that off-label iNO use is detrimental.
Reply:
Thank you for your instructive advice. Recently, there have been more and more studies to investigate the efficacy of iNO in preterm neonates with PPHN or severe RDS. Please see the references no.8~12. These studies were conducted on preterm infants who received iNO, and improved pulmonary outcomes have been documented. Please also see reference no. 31 and no. 32, American Heart Association, American Thoracic Society and Pediatric Pulmonary Hypertension Network have considered iNO to be superior to other pulmonary vasodilators in the treatment of preterm infants with severe pulmonary hypertension.
The authors have to define iNO rescue therapy in the manuscript.
Reply:
Thank you for your instructive advice. I will revise the term “iNO rescue therapy” to be “off-label use of iNO as a rescue therapy”. I will not use “iNO rescue therapy” in the revised manuscript. Please also note that I classified the use of iNO as standard use (for PPHN, irrespective of the underlying etiology) and the off label use (for clinically presumed pulmonary hypertension or diagnoses other than PPHN) (Page 6, the first paragraph, and from the third line).
In this study, we aimed to test the effectiveness and safety of iNO off-label use in neonates without definite diagnoses of PPHN – inTaiwan, was iNO use deemed off-label?
Reply:
Thank you for your question. InTaiwan, iNO use was not deemed off-label. We used iNO as standard therapy for near-term and term-born infants with documented PPHN, and iNO off-label use is applied in preterm infants with PPHN and sometimes in patients with secondary pulmonary hypertension and poor response to other therapeutic strategies.
Materials & methods: The iNO practical policy in the NICUs of Linkou CGMH included the standard rescue therapy for refractory hypoxemia due to PPHN, which was documented by echocardiography reports or 88 oxygen saturation differences of > 5-10% or PaO2 differences of 10-20 mmHg between lower limbs and right upper limb before initiation of iNO.
The median (IQR) median duration of iNO use was 5.0 (3.0-11.0) days – kindly explain –how did this timeframe include infants with BPD?
Reply:
Thank you for your question. The timeframe is not when the patients had iNO therapy. Instead, it indicates the duration of iNO use. Therefore, it is possible an infant with BPD, maybe at PCA 36 weeks, had received iNO for five days due to refractory respiratory failure.
However, the most common diagnosis of neonates with iNO off-label use after one week old were severe BPD (70.4%) and secondary pulmonary hypertension (51.9%) – Please explain. This is contradicting the above statement.
Reply:
Thank you for your question. Of course these subjects were only clinically presumed pulmonary hypertension and did not have the definite diagnosis of PPHN. However, iNO is sometimes used as a final rescue therapy for these patients with secondary pulmonary hypertension due to severe BPD. Therefore, these patients had “Off-label use of iNO”, and the objective of this study is to investigate these usages.
Besides, the control group had comparable gestational age and birth body weight with those in the early use of iNO off-label use group. In a retrospective study, what does comparable mean? Comparable is not a scientific term in a retrospective study and make the study weak.
Reply:
Thank you for your question. Because we aimed to compare the safety, efficacy, and outcomes of neonates who received iNO (the study group) with those who did not receive iNO (the control group), it is better that these two groups had comparable gestational age (GA) and birth weight (BW). In NICU studies, the demographics of GA and BW are inevitably the confounding factors that will affect the outcomes, since various neonatal comorbidities are significantly more common in extremely preterm infants and low birth body weight infants.
Neonates with iNO use for PPHN or those with off label use of iNO had significantly more severe hypoxemia, higher oxygenation index, higher AaDO2, higher mean airway pressure and more hypercabnia when compared with the control group (all P values < 0.05). – American Thoracic Society recommends that CO2 levels normalize before iNO initiation. Kindly comment.
Reply:
Thank you for your instructive advice. In our institute, the use of iNO depends on attending physician’s decision. We usually use iNO as the final rescue, or in the most severely ill neonates. Therefore, neonates with iNO use for PPHN or those with off label use of iNO had significantly more severe hypoxemia, higher OI, higher AaDO2, higher MAP and more hypercabnia than the controls. I have mentioned this issue in the first paragraph of page 14, the discussion section. I did not find American Thoracic Society to recommend that CO2 levels normalize before iNO initiation.
In neonates with iNO use, more than 70% of patients had an OI greater than 20 at initiation of iNO, and more than 75% of patients had initiation of iNO within 48 hours after initiation of HFOV, which indicated that these patients did not respond to HFOV under the maximum settings. – How will they respond if there exists respiratory acidosis with hybercarbia?
Reply:
Thank you for your question. For neonates with respiratory acidosis with hypercarbia, the HFOV will be set to wash out CO2 and correct the hypercarbia. Once hypercarbia is corrected, the respiratory acidosis can be corrected.
The authors cited some papers – response to iNO is not merely change in FiO2 (expressed as percentage not decimal), some papers have mentioned it as fraction of change in PaO2 to FiO2.
Reply:
Thank you for your instructive advice. I agree with reviewer’s comments that the response to iNO and HFOV can also be based on fraction of change in PaO2 to FiO2. However, most studies used reduction of FiO2 after initiation of iNO and HFOV as judgment of therapeutic response. I will cite some papers in the study outcome section, page 8, references no. 10,12,22.
Discussion needs to be focused. Was years of use added as a variable?
It is a good description of the data of the patients about the NO use, which has been controversial in refractory hypoxemia in the absence of PPHN.
Reply:
Thank you for your instructive advice. Years of use has been added as a variable, but does not contribute to significant difference because the treatment outcomes did not changed during the study period. For discussion section, I focused on the main findings, then the therapeutic response, the factors of therapeutic response, and then the limitations and strengths of this study. I appreciate your suggestions and comments. Would you please point out more concisely which I should do to improve the discussion
Reviewer 2 Report
It is a good description of the data of the patients about the NO use, which has been controversial in refractory hypoxemia in the absence of PPHN.
- Some sentences need to be rewritten for grammatical errors example 1. In Objectives of the abstract use the aim of the study rather and " we aimed", 2. Result section of abstract: use the word worse rather than " significantly worse"
-What was the percentage of rescue use of NO in your study for the patients? Also, can you describe the rescue use of NO percentage with gestational age in your patient population group?
- Can you describe why you chose case-control as the design of the study rather than writing the data as a descriptive study?
- What was the rate of IVH in your patient population which received the rescue NO?
- Rescue use of NO in the past studies has not shown any improvement in survival before, similar to your study even though it has the potential to transiently and successfully improve the oxygenation.
How does this study add to the literature?
- Sick pre-term patients in which ended up getting the rescue NO had high mortality. What is the significance of the multivariate analysis described? Are these results different from the multivariate analysis of the sick patient population only, without the use of NO being a factor?
- Conclusion: needs to be re-written. This study does not justify the cost-benefit analysis to use NO in refractory hypoxemia given high mortality. Perhaps it will be more reasonable to rephrase that NO continues to be used as an off label rescue in refractory hypoxemia.
Author Response
RE: JCM-524850
Rescue off label use of inhaled nitric oxide in neonates with refractory hypoxemic respiratory failure: therapeutic response and risk factors of mortality
Journal of Clinical Medicine
Dear Editor,
Thank you for your appreciated comments on our manuscript. We had the manuscript revised, all according to the reviewers’ and editor’s suggestions. We underline every change and highlight in red color on the revised manuscript. The replies for the reviewers’ criticisms are as followings. We hope this revised version can be acceptable.
Best regards,
Ming-Horng Tsai
Chief, Division of Neonatology and Pediatric Hematology/Oncology, Department of Pediatrics, Yunlin Chang Gung Memorial Hospital,Taiwan, R.O.C.
Comments from Reviewer No.2 :
- Some sentences need to be rewritten for grammatical errors example 1. In Objectives of the abstract use the aim of the study rather and " we aimed", 2. Result section of abstract: use the word worse rather than " significantly worse"
Reply:
Thank you for your instructive advice. I will request professional English editing from specialized company and native English speaker. The revised manuscript has been corrected after professional English editing. All these errors will be corrected in the revised manuscript, thank you.
- What was the percentage of rescue use of NO in your study for the patients? Also, can you describe the rescue use of NO percentage with gestational age in your patient population group?
Reply:
Thank you for your instructive advice. In the revised manuscript, I will not use the term “rescue use of iNO”. Instead, I will use off-label use of iNO as a rescue therapy. Please note that the title is also revised. In our institute, use of iNO can be the standard use or off-label use, please see the definition section, page 6, the first paragraph. It is difficult to define “rescue use of iNO” because sometimes other pulmonary vasodilators can also be used, but the clinicians did not try. All these cases have been enrolled. For the question, I think I can answer: For all neonates with PPHN, 122/206 (59.2%) have received iNO as the rescue therapy. (I will add this sentence in the result section, page 9, 2nd paragraph, the 10th line). In my patient population group, the use of iNO percentage with gestational age can be described, but this will make the results or the table 1 too long. Since the median gestational age of the study group has been listed in Table 1, and I have a description of these demographics in the first paragraph of result section, page 9. Please inform me if you do insist to have these data in the revised manuscript.
- Can you describe why you chose case-control as the design of the study rather than writing the data as a descriptive study?
Reply:
Thank you for your question. Because we aimed to investigate the therapeutic benefit of iNO as a rescue therapy in the most critically ill neonates in the NICU, the control group with PPHN or refractory respiratory failure but did not receive iNO is required. Then we can conclude the therapeutic impacts, influences and beneficial effects of iNO as a rescue therapy. If only a descriptive study is conducted, we will not know the effects of iNO if other pulmonary vasodilators were prescribed.
- What was the rate of IVH in your patient population which received the rescue NO?
Reply:
Thank you for your question. The percentage of IVH in our patient population which received the rescue NO was 9.2% (19 out of 206 patients). I will add the sentence in the result section: The percentage of IVH (≥ grade III) in infants who received iNO therapy was 9.2% (19/206), which was comparable to that in the control group (8.0%). (page 9 of result section, last 5th line)
- Rescue use of NO in the past studies has not shown any improvement in survival before, similar to your study even though it has the potential to transiently and successfully improve the oxygenation.
How does this study add to the literature?
Reply:
Thank you for your question. Although we did not prove rescue iNO can improve the survival, we did document that rescue iNO is a feasible therapeutic strategy that can improve the oxygenation. Besides, in the population of refractory respiratory failure, the most severe respiratory failure and highest OI were not the independent risk factor for final mortality. Instead, extremely preterm, severe hypotension, and pulmonary hemorrhage were independently associated final mortality after multivariate logistic regression. This means we can improve the survival of these critically ill patients once we can solve the problems of severe hypotension and pulmonary hemorrhage. I think this is what we add to the literature. Please also see “what this study adds” section in page 19, thank you.
- Sick pre-term patients in which ended up getting the rescue NO had high mortality. What is the significance of the multivariate analysis described? Are these results different from the multivariate analysis of the sick patient population only, without the use of NO being a factor?
Reply:
Thank you for your question. The multivariate analysis shows the independent risk factors for mortality are severe hypotension and pulmonary hemorrhage, rather than more severe hypoxemia (higher OI). This indicated that iNO can reverse initial severe hypoxemia and we can improve the survival of these critically ill patients if we can solve the problems of severe hypotension and pulmonary hemorrhage.
It is unknown whether these results will be different from the sick patient population only without the use of iNO being a factor. Because the multivariate analysis was conducted in all neonates with iNO therapy, that means all sick patients in the population had received iNO therapy, it seems impossible to exclude use of iNO being a factor.
- Conclusion: needs to be re-written. This study does not justify the cost-benefit analysis to use NO in refractory hypoxemia given high mortality. Perhaps it will be more reasonable to rephrase that NO continues to be used as an off label rescue in refractory hypoxemia.
Reply:
Thank you for your instructive advice. I will rephrase the conclusion as the following: iNO is applicable to be an off-label rescue therapy for premature neonates with refractory hypoxemia due to severe pulmonary hypertension and bronchopulmonary dysplasia. (The conclusion in the abstract)